# ACME++: Secure ACME Client Verification for Web-PKI

Paper # 2631, 8 pages body, 9 pages total

## Abstract

The Automatic Certificate Management Environment (ACME) protocol automates the SSL certificate issuance and renewal process, streamlining large-scale certificate management. Its caching mechanism allows Certificate Authorities (CAs) to store domain validation results for up to 30 days. While this mechanism reduces the burden of re-validation on the CA server, it also introduces a vulnerability where attackers can bypass domain validation using stolen ACME account credentials.

In this paper, we introduce the ACME *Authz* Cache Attack, a method that enables attackers to request fraudulent certificates without domain control. We demonstrate that even Let's Encrypt, the world's largest CA, is susceptible to this vulnerability. To address this issue, we propose ACME++, an enhanced version of the protocol that binds the ACME account to the client's IP address and a unique ID, ensuring re-validation for each certificate request from a new client. Our implementation of ACME++ shows that it effectively mitigates this attack with minimal impact on server performance.

## 1 Introduction

Today, about 96% of web traffic through Google is secured via HTTPS [22], ensuring data integrity and privacy for online services such as e-commerce and video streaming. At the core of these secure connections are SSL certificates, which are issued and managed under the Web Public Key Infrastructure (Web-PKI). In the Web-PKI ecosystem, CA authenticates websites by binding domain identities to public cryptographic keys via SSL certificates. During the HTTPS connection establishment, the web server presents the certificate to the client for secure communication verification.

For a single web server, requesting and deploying an SSL certificate involves minimal cost and effort. However, managing SSL certificates at scale in large enterprises introduces considerable complexity. Manually requesting, deploying, and renewing certificates becomes time-consuming and error-prone, and even a single expired or misconfigured certificate can lead to service outages and substantial financial losses [3, 35].

To address these challenges, the ACME protocol [4] was introduced and has been widely adopted by major CAs, automating certificate lifecycle management tasks such as issuance, renewal, and revocation. One notable example is Let's Encrypt [26], a CA that issues certificates exclusively through ACME. By 2021, Let's Encrypt accounted for nearly 30% of certificates found in IPv4 scans and over 80% of those in CT logs [18], playing a key role in driving the adoption of ACME-based certificate issuance. Other commercial CAs have also integrated the ACME protocol into their platforms [15], making ACME a standard across the industry.

Despite its widespread adoption and operational efficiencies, ACME introduces certain vulnerabilities into the Web-PKI ecosystem. A key vulnerability stems from ACME's caching of domain validation records, a mechanism intended to reduce the workload on CA servers that can be exploited by attackers to bypass domain re-validation and obtain fraudulent certificates. Such certificates enable attackers to conduct man-in-the-middle (MITM) attacks. For instance, in 2011, hackers breached the CA server of DigiNotar [19] and issued fraudulent certificates, enabling them to intercept user traffic intended for Google services [28]. Similarly, in 2015, Google uncovered that MCS Holdings had deployed self-issued fraudulent certificates in proxy devices designed to intercept secure traffic [6]. These incidents illustrate how vulnerabilities—whether from operational errors or design flaws like those in ACME—can be exploited, compromising user data and privacy and posing significant risks to the integrity of the Web-PKI ecosystem.

In this paper, we introduce the ACME *Authz* Cache Attack, a method that exploits ACME's caching mechanism to obtain fraudulent certificates without domain control. We show that attackers, after accessing a victim's ACME account credentials, can bypass domain re-validation by reusing cached authorization (*Authz*) records. This attack can be carried out with a standard internet connection without complex network manipulation. Moreover, attackers can broaden their targets by identifying associated domains through CT logs, some of which may have cached *Authz* records under the same ACME account. Our evaluation reveals that Let's Encrypt is vulnerable to this attack, demonstrating its widespread impact.

To counter this threat, we propose ACME++, an enhanced version of the ACME protocol that strengthens security by binding client IP addresses and unique identifiers to ACME accounts while requiring domain re-validation for each certificate request from a new client. ACME++ is designed for seamless integration with the existing ACME framework, reusing current objects and message exchange standards to minimize protocol modifications. We implemented ACME++ on both the ACME client and CA server, demonstrating its resilience against similar attacks while maintaining operational efficiency on the CA server.

The main contributions of this paper are as follows:

- We identify and analyze the vulnerabilities in ACME's caching mechanism, focusing on how the reuse of cached *Authz* records can be exploited by attackers. Additionally, we highlight ACME implementation flaws in the use of incrementally generated account IDs, which further weaken the security of the Web-PKI ecosystem and increase the risk of unauthorized issuance.

- We introduced the ACME *Authz* Cache Attack, a novel attack in which an attacker can obtain fraudulent certificates without domain re-validation, once the attacker acquires ACME account credentials. We also demonstrate that the attacker can leverage CT logs to identify associated domains, with an expectation that about half of these domains are valuable for targeting.

- We designed ACME++, an enhanced version of ACME, to mitigate this attack. ACME++ incorporates additional security checks for ACME clients communicating with CAs, ensuring that stolen credentials cannot be exploited. We evaluate ACME++ and show that it introduces no more than a 50% increase in traffic and a 60% increase in time overhead for CAs in worst-case scenarios while proving its resilience against other potential attack vectors.

The remainder of this paper is organized as follows: Section 2 provides the background on the ACME protocol and its vulnerabilities. Section 3 details the ACME *Authz* cache attack, while Section 4 outlines our proposed mitigation, ACME++. We discuss other related works for Web-PKI in Section 5, and Section 6 concludes the paper.

## 2 Background

In this section, we introduce the foundational concepts related to attack vectors and mitigation strategies. We begin with an overview of the Web-PKI ecosystem (Section 2.1), focusing on its role, structure and SSL certificates. Next, we delve into the ACME protocol and its caching mechanisms (Section 2.2). Finally, we highlight vulnerabilities within the implementation of the ACME protocol (Section 2.3), which serve as the basis for our discussion on potential attacks and corresponding mitigation techniques.

### 2.1 Web-PKI and SSL Certificates

Web-PKI is a public key infrastructure (PKI) designed to secure communication between websites and users, preventing data tampering and MITM attacks. As noted in [13], the core entities in a PKI include CAs and End Entities (EEs). PKI leverages digital certificates to authenticate encrypted communication between these entities. For a more detailed analysis of PKI architecture, readers can refer to [10, 11, 31].

The CA plays a central role in the Web-PKI framework, managing certificates for all EEs. Web servers act as EEs by deploying SSL certificates to establish secure HTTPS connections with the client. As cryptographic assertions, SSL certificates bind the *subject*, typically a domain name, to the web server's *public key*. In accordance with the X.509 standard [13], SSL certificates also contain fields such as validity periods, issuer, and X.509 extensions. These certificates are issued and signed by trusted CAs, which themselves hold CA certificates. The CA's certificate is signed by a higher-level CA, ultimately forming a "chain of trust" that terminates at a trusted root CA certificate. When users access a website, their browsers verify this certificate chain to ensure the website's authenticity. Through this design, Web-PKI enables users to authenticate web servers, ensuring the privacy and integrity of data transmissions.

### 2.2 ACME Protocol with Cache Mechanism

The ACME protocol [4] was developed to mitigate the challenges and inefficiencies inherent in manual SSL certificate management for large-scale systems. The manual processes of deploying, renewing, and replacing SSL certificates across numerous servers are prone to human error, time-intensive, and frequently result in misconfigurations or certificate expirations.

ACME automates the entire certificate lifecycle—including request, issuance, renewal, and revocation. For each of these actions, the protocol specifies a unique URL. The client must send a *request* to the corresponding *URL* to complete the desired *action*. CA servers provide clients with a Directory object, allowing them to configure the correct URLs for each ACME action. Table 1 lists all available services and their corresponding URLs in the CA Directory. The Directory is a JSON object with field names sourced from the resource registry and values as the corresponding URLs. In addition

**Table 1: ACME CA Directory Services**

| Action | Directory URL | Service Description |
|---|---|---|
| *newNounce* | /acme/new-nonce | Getting a new nonce from CA to avoid replay attack. |
| *newAccount* | /acme/new-account | Registering a ACME account. |
| *newOrder* | /acme/new-order | Requesting a new certificate order from the CA. |
| *newAuthz* | /acme/new-authz | Creating a *Authz* object for pre-authorization (Optional). |
| *revokeCert* | /acme/revoke-cert | Submitting a revocation request for a certificate. |
| *keyChange* | /acme/key-change | Changing the public key binding to an account. |

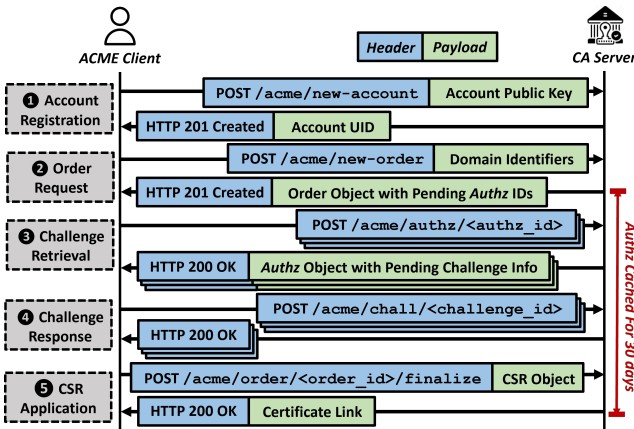

**Figure 1: Certificate Request Process via ACME Protocol. Valid *Authz* records are cached for 30 days after the CA validates the client's domain ownership.**

to the Directory, the CA also provides resource URLs for the client to retrieve objects. For instance, the client can access the corresponding *Authz* record through /acme/authz/<authz_id>, where <authz_id> is the unique identifier of the object.

Today, many CAs have integrated ACME into their operations. Let's Encrypt [26], which was the first CA to exclusively issue billions of free Domain Validation (DV) certificates using ACME, serves as a prominent example. Through ACME clients [1, 16], website operators can easily obtain certificates with a 90-day lifespan. Figure 1 outlines the typical process of requesting an SSL certificate using the ACME protocol. In the initial phase, the client submits a *newAccount* request to the CA, providing a signed payload containing an email address and public key. For non-Let's Encrypt CAs, external account binding (EAB) requires both an HMAC key and a key identifier (KID). Once verified, the CA creates an account and returns a unique account URL with an associated ACME account ID. The client then sends a *newOrder* request specifying the domains for which the certificate is requested.

To validate domain ownership, the CA issues one or more challenges (e.g., HTTP-01 or DNS-01 [25]). Upon successful completion of these challenges, the CA caches the domain validation result, referred to as an authorization object, *Authz*, for 30 days. This

caching allows quicker reissuance of certificates for the same domains without requiring domain re-validation. When the client requests a new certificate for previously validated domains, it submits a *newOrder* request as usual. If the *Authz* records for these domains remain valid, the client can bypass the re-validation step and directly submit a Certificate Signing Request (CSR) to obtain the certificate.

However, this caching mechanism presents a critical security vulnerability. As we demonstrate in Section 3, an attacker can exploit the cached *Authz* to acquire fraudulent certificates using the victim's ACME account credentials without the need to manipulate network traffic. Under normal operational conditions, the CA would fail to detect such an attack.

### 2.3 ACME Account Implementation Flaw

In theory, the ACME account ID included in the URL should be randomly generated to prevent unauthorized access. The RFC recommends that ACME account unique identifiers (UIDs) be cryptographically secure. However, in Let's Encrypt's implementation, account IDs are assigned as incrementally generated integers, which weakens the system's resilience. Attackers can infer valid UIDs based on the pattern of server responses or by brute-forcing the range of possible account IDs. Once attackers discover a valid UID, they can exploit the system further by using the victim's account and leveraging cached *Authz* to request fraudulent certificates.

In Section 3, we demonstrate how this vulnerability leads to unauthorized access and misuse of ACME accounts. By exploiting the predictable account UIDs and cached domain *Authz* records, attackers can successfully issue fraudulent certificates without the need to manipulate network traffic. This flaw not only undermines the integrity of the certificate issuance process but also poses a significant threat to the security of the broader Web-PKI ecosystem, potentially facilitating MITM attacks, phishing campaigns, and other malicious activities.

## 3 ACME *Authz* Cache Attack

We introduce a novel attack exploiting the ACME protocol's caching mechanism to acquire fraudulent certificates, based on the observation that cached *Authz* records can be considered as the ownership credentials of the domains. We first outline the threat model (Section 3.1), followed by a detailed description of the attack methodology (Section 3.2). Finally, we present real-world experiments, demonstrating the vulnerability of Let's Encrypt to this attack (Section 3.3), and discovering the number of associated domains for target websites (Section 3.4).

### 3.1 Threat Model

The attack objective is to obtain a fraudulent SSL certificate containing as many victim domains as possible, enabling the attacker to impersonate the victim's server through the MITM attack.

**Attack Assumptions:**
We assume the following conditions for the attacker and victim:

- The victim's server stores its ACME account private key and other ACME account credentials locally on the server.

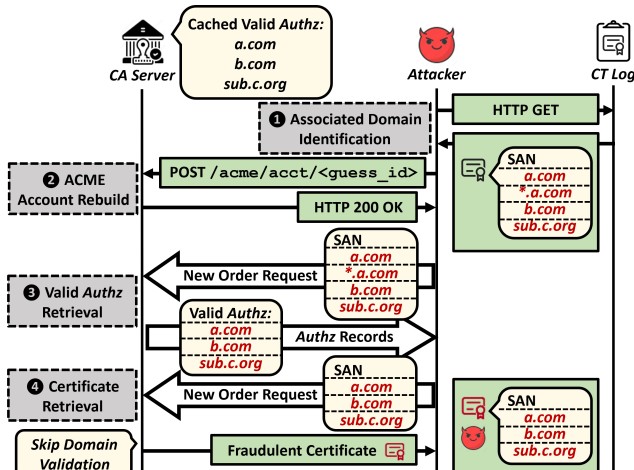

**Figure 2: ACME *Authz* Cache Attack with Four Steps. The ACME Account Rebuild step can be skipped if the attacker has obtained full credentials of the victim's ACME account.**

- The victim has requested certificates in the last 30 days and the corresponding *Authz* records are cached by the CA and available for reuse.
- The attacker obtains the victim's ACME account credentials by exploiting a server-side vulnerability. One plausible vector is the Path Traversal Vulnerability [29], which allows unauthorized access to files outside intended directories through server misconfiguration [23]. This vulnerability enables the attacker to access ACME account credentials.
- The attacker knows at least one domain service deployed on the victim's server.

**Attacker Abilities:**
We consider an attacker with minimal capabilities. While previous threat models on Web-PKI involve more complex capabilities such as compromising an Autonomous System border router or manipulating the traffic on CA DNS resolvers or nameservers [5, 8, 14], in our threat model, the attacker only has a standard Internet connection that can scan web servers to uncover vulnerabilities and domain services.

### 3.2 Attack Methodology

Figure 2 illustrates this attack through an example. The attack consists of four main steps: identifying associated domains, rebuilding the victim's ACME account profile, retrieving valid *Authz* records, and getting fraudulent certificates. In this example, the victim has an ACME account and requested certificates for a.com, b.com, and sub.c.org. These domain *Authz* records remain valid during the attack. The attacker knows a.com as the target domain.

*3.2.1 Associated Domain Identification.* The first step is to identify as many associated domains as possible. A domain (including wildcard domain) is called an *associated domain* if it appears in the same SSL certificate as the target website domain. Website operators frequently group multiple domains hosted on the same server in a single certificate to reduce operational costs and complexity.

Through ACME protocol, *Authz* records for these associated domains and target domains lie under the same ACME account, making them high-value targets for attackers aiming to expand their attack surface. To discover associated domains, attackers can search through certificates archived in public CT [21] logs. Although CT logs contribute to fraudulent certificate detection at the earliest stage, they may also leak sensitive information to third parties, such as Fully Qualified Domain Names (FQDNs) and business relationships [30, 32, 33]. In this attack, the attacker can identify certificates issued for the target domain in the last 30 days, and find associated domains in the certificates' Subject Alternative Name (SAN) extension. In this example, the associated domains of the target domain `a.com` are `*.a.com`, `www.a.com`, `b.com`, and `sub.c.org`.

*3.2.2 ACME Account Rebuild.* In the second step, the attacker reconstructs the victim's ACME account profile from the obtained ACME account credentials. If the attacker knows both the ACME account UID and private key (full credentials), the account profile can be easily restored by setting the correct storage path. If the attacker only obtains the private key, the account UID can be determined by a brute-force method from CA ACME implementation flaws discussed in Section 2.3. The attacker sends a POST-as-GET request to `//acme/acct/<guess_id>` with an integer UID to the CA server. If the UID matches the victim's ACME account private key, the server responds with an HTTP 200 OK, returning the account object. If not, an error is returned. By iteratively adjusting the UID, the attacker can eventually uncover the correct ACME account profile.

*3.2.3 Valid Authz Record Retrieval.* Next, with the account profile and associated domains, the attacker tries to discover valid *Authz* records. The attacker submits a certificate request containing all discovered domains. The CA returns the order object with all *Authz* URLs. The attacker can then request to `/acme/authz/<authz_id>` directory with the corresponding ID to check the *Authz* status, allowing the attacker to distinguish between domains with active cached authorizations and those without. In the example, the order request consists of `a.com`, `*.a.com`, `www.a.com`, `b.com`, and `sub.c.org`. Only `a.com`, `b.com`, and `sub.c.org` have valid records, while `*.a.com`, `www.a.com` has record status of "pending".

*3.2.4 Fraudulent Certificate Retrieval.* In the last step, the attacker requests a fraudulent certificate with the valid domains. As the *Authz* records for these domains are all valid, the client can bypass undergoing domain control challenges and directly obtain the fraudulent certificate. [1] In our example, the attacker discovers valid *Authz* records for `a.com`, `b.com`, and `sub.c.org`, and successfully obtains a fraudulent certificate for these domains.

## 3.3 Evaluation Against Let's Encrypt

*3.3.1 Setup.* We evaluated the proposed attack against Let's Encrypt, the largest CA fully utilizing the ACME protocol. The victim's server hosts the target domain `sparklestar.work` and resides on a separate network from the remote attacker. The victim uses Certbot [16] as the ACME client to request certificates. Specifically,

the victim requests certificates for the domains `sparklestar.work` and `*.sparklestar.work`, with ACME account credentials stored locally on the server. To simulate the attack, we configure the victim's server to be vulnerable to a path traversal vulnerability due to Nginx alias misconfiguration [23], which the attacker exploits via internet scanning.

*3.3.2 Requesting Fraudulent Certificates.* Through the path traversal vulnerability, the attacker acquired the victim's full ACME account credentials, including the account UID and private key. Following the attack procedure, the attacker first located the victim's certificate in Google's CT log (argon2024 [20]) and identified associated domains (`sparklestar.work` and `*.sparklestar.work`). The attacker then used Certbot to submit a new certificate request to Let's Encrypt. Due to the cached *Authz* records on the CA's ACME server, the attacker successfully obtained a certificate for the victim's domains without undergoing any domain validation. This confirms the basic feasibility of the attack.

*3.3.3 Adding New Domains.* We extended our evaluation to examine whether an attacker could append additional domains, such as those intended for phishing attacks, to the fraudulent certificate. In such scenarios, when a web user connects to a phishing website and inspects the certificate, it becomes more challenging to distinguish malicious domains, as they appear alongside legitimate ones. For this experiment, the attacker registered a new domain `sparkle-star.org` and followed the procedure outlined in 3.3.2. This time, the attacker requested a certificate with `sparkle-star.org` included in the SAN field. As anticipated, Let's Encrypt only required domain validation for the newly added domain `sparkle-star.org`, bypassing validation for `sparklestar.work` and `*.sparklestar.work`, since their *Authz* records remained valid in the cache.

As a result, the attacker successfully obtained a certificate that includes `sparkle-star.org`, `sparklestar.work`, and `*.sparklestar.work`. This experiment demonstrates a serious security vulnerability, as attackers can not only obtain fraudulent certificates for legitimate victim domains but also append unrelated or malicious domains. Such behavior further erodes the trustworthiness of the Web-PKI infrastructure by potentially allowing fraudulent domains to appear as trustworthy to unsuspecting users.

*3.3.4 Let's Encrypt ACME Implementation Flaws.* In the previous scenario (Section 3.3.2), the attacker had all account credentials, without the need to rebuild the account. We also investigated scenarios where the attacker only possesses the victim's ACME account private key and attempts to obtain fraudulent certificates. As detailed in Section 3.2.2, the attacker must determine the account UID to complete the ACME account profile. We used a script to simulate a brute-force approach, continuously increasing the account integer UID. Our results showed that once the account UID matches the private key, the attacker can successfully rebuild the victim's ACME account and proceed to request certificates. This indicates that Let's Encrypt's ACME implementation has a critical flaw in generating account UIDs.

## 3.4 Associated Domain Analysis

We evaluated the attack surface by analyzing the discovery of associated domains, aiming to determine how many associated domains

---

[1]In fact, the attacker can obtain more than one fraudulent certificate. The number of certificates an attacker can request is limited by the CA's rate limits. For Let's Encrypt, the limit is 50 certificates per registered domain per week [17].

**Table 2: Top-1M Websites Associated Domain Data Overview. Certificates are issued in Sept. 2024.**

| CT Log Name | Nimbus [12] | Sabre [34] |
|---|---|---|
| #Certificates | 1,195,862 | 663,813 |
| #Top Domains in Certificates | 135,569 | 117,993 |
| #Associated Domains | 7,314,093 | 1,784,209 |
| #Domains with JARM FP | 842,671 | 525,596 |

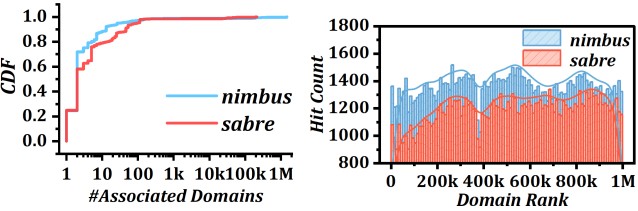

**(a) CDF Graph For #Associated Domains (In Log Scale).**

**(b) Websites Rank Distribution (Bin = 10,000).**

**Figure 3: Associated Domains Distribution Result.**

could be identified given a target website. Using domains from Cisco's Top-1M websites [36] as test targets, we searched for certificates containing associated domains through CT logs. Each CT log was analyzed independently to simulate a scenario where an attacker might focus on a specific log. For each target domain, we retrieved certificates issued in September 2024 from these logs and enumerated the associated domains.

As CT logs do not directly indicate where certificates are deployed, simply extracting certificate copies can produce many associated domains, including false positives. To refine the analysis, we verified associated domains using JARM TLS fingerprinting [2]. Domains sharing the same JARM fingerprint suggest that the web servers for these domains are managed by the same administrator or organization. Administrators using different ACME accounts may encounter re-validation of domains, leading them to use the same ACME account for convenience. As a result, certificate requests for these domains originate from a single ACME account, with valid *Authz* records cached under that account. This means that compromising the management account for one domain could potentially be used to attack many other associated domains.

The results are summarized in Table 2. Preliminary results indicate that 135,569 and 117,993 target domains have certificates and associated domains in Nimbus and Sabre, respectively. We first calculated the number of associated domains for these target domains. As illustrated in Figure 3a, over 40% of these target domains have at least one associated domain beyond themselves in Sabre, and about 30% in Nimbus. The CDF line shows a long tail with an extremely large number of associated domains. Certain target domains (e.g., domains owned by CDNs) have over 1 million associated domains, presenting a substantial attack surface.

We then analyzed the relationship between the presence of associated domains and the rank of the target websites. Figure 3b displays the distribution of these domains in Nimbus and Sabre, showing similar distributions across the rank spectrum. This suggests that attackers can identify potential targets both among highly ranked and less prominent domains.

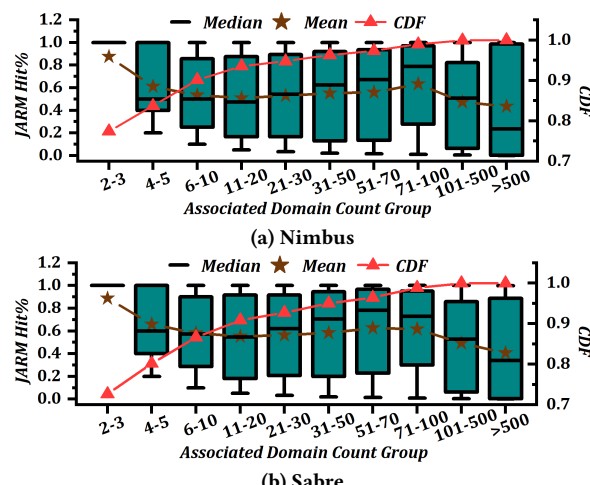

**(a) Nimbus**

**(b) Sabre**

**Figure 4: Associated Domain JARM Hit Percentage In Target Domain Groups and CDF Line for Group Domain Counting in Different CT Logs.**

Figure 4 presents the JARM hit results. The x-axis represents the group index, where each group corresponds to a specific range in the number of associated domains. The CDF line represents the cumulative percentage of domains within each group, while the boxplots display the JARM hit rate percentage for each target domain group. Group lengths were selected to ensure a smooth increase in the CDF line.

The results indicate that across all groups in Nimbus (Figure 4a) and Sabre (Figure 4b), the median JARM hit rates are consistently above 50% (except the last group), suggesting that the expected proportion of true associated domains identified in each CT log exceeds 50%. Although the last group exhibits higher variability and a lower median, this group comprises a very small proportion of target domains (<0.01%), thus having minimal impact on the overall findings. Furthermore, for almost all target domains in the first group, the JARM hit rate reaches 100%, indicating a high probability that these associated domains are hosted on the same server, making them susceptible to potential attacks. Furthermore,

**Takeaways.** In conclusion, the associated domain analysis demonstrates that an attacker could identify numerous target domains across a broad range of ranks in each CT log, with associated domain counts ranging from 1 to more than 1,000,000. Approximately half of these associated domains are likely to be hosted on the same server, making them valuable targets in the third step of the ACME *Authz* Cache Attack in Section 3.2.3.

## 3.5 Ethics Considerations

This research investigates vulnerabilities in the ACME protocol and its implementation by major CAs, with a focus on how malicious actors can exploit these weaknesses to obtain fraudulent SSL certificates. The primary goal is to improve the security of the Web-PKI ecosystem and raise awareness of risks related to ACME misconfiguration. All experiments involving Let's Encrypt were conducted in a controlled environment, utilizing test domains that we registered and servers intentionally configured with known vulnerabilities for evaluation purposes. For the domain analysis,

all certificate data was sourced from public CT logs. When getting JARM fingerprinting, only 10 Client Hello messages were sent to each target server per scan, ensuring minimal risk of triggering denial-of-service attacks.

## 4 ACME++: Secure ACME Client Verification

The core vulnerability in our ACME *Authz* Cache Attack stems from a design flaw in the ACME protocol, which treats possession of an ACME account as sufficient proof of domain control. To mitigate this threat, we propose **ACME++**, an enhanced version of the ACME protocol that introduces additional verification mechanisms for account ownership while preserving the existing Web-PKI infrastructure and compatibility with current CA configurations. In this section, we detail the design principles (4.1 - 4.3), and experimental evaluation of ACME++ (4.4).

### 4.1 Design Goals

ACME++ builds upon the existing cache mechanisms in ACME, incorporating supplementary checks for clients sharing the same ACME account. The design of ACME++ adheres to the following key objectives:

- **Minimal overhead for CAs.** The original ACME cache mechanism was introduced to reduce the computational burden on CAs by avoiding redundant domain control validation during high-frequent certificate reissuance. ACME++ follows this principle, ensuring that additional verification steps are lightweight and leverage caching to minimize CA overhead.
- **Ease of deployment.** ACME++ is designed as a seamless upgrade to existing ACME implementations, requiring minimal modifications to both the client and CA server. The design remains aligned with the original ACME CA server structure and protocol transmission process, adding new functionalities within the existing CA server framework without introducing entirely new methods.
- **Robust security.** ACME++ must not introduce new attack vectors into the Web-PKI system, such as vulnerabilities to brute-force attacks. It must strengthen security while maintaining resilience against existing and potential future threats.

### 4.2 *Client Authz* Object Design

The key improvement introduced in ACME++ over the original ACME protocol is the addition of enhanced verification mechanisms for ACME clients, specifically aimed at preventing attackers from obtaining fraudulent certificates after compromising a victim's ACME account private key. Building on the core principles of *Authz* and domain control challenges, ACME++ introduces *Client Authz*, a novel data object for client verification.

The *Client Authz* object (format detailed in Table 3) serves as a unique verification token that binds an ACME client to a specific account. Each object associates a unique `identifier` with both the client's IP address and a client-specific ID. The IP address corresponds to the machine executing the ACME client, while the client ID is a string generated by the client. This identifier scheme not only helps to differentiate legitimate ACME clients from potential attackers but also enhances resilience against brute-force attempts and similar attack vectors (further discussed in Section 4.5). Similar

**Table 3: *Client Authz* Object Fields**

| Field Name | Contents |
|---|---|
| status | Current verification status of the `identifier`. |
| expires | Current verification expiration time. |
| identifier | The unique identifier of the client to be verified. It consists of the client IP and a client ID generated randomly. |
| challenges | A list of ACME challenge objects for client verification, each corresponds to `identifier`, `type`, `url` and `toekn`. |

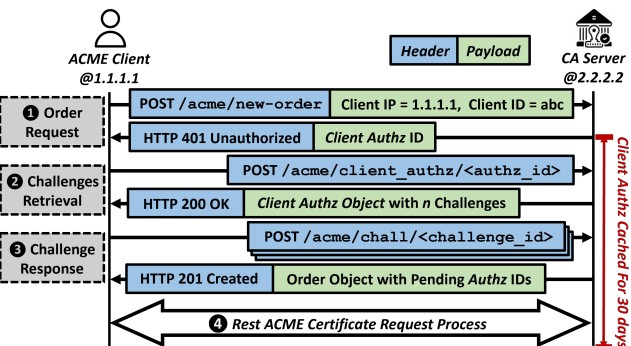

**Figure 5: ACME++ Flow Illustration and Message Change**

to the domain *Authz* record in ACME, the CA caches the Client *Authz* object for 30 days, provided that the associated `challenges` are completed. The `challenges` field contains multiple Challenge objects that must be fulfilled by the client. The specific method for generating these challenges will be described in Section 4.3.

### 4.3 Protocol Design

The ACME++ workflow is illustrated in Figure 5. When an ACME client initiates a *newOrder* request, it includes its IP address and client ID along with its ACME account key and UID in the request to the CA server. Upon receiving the request, the CA performs two verification steps: (1) checking if the client's communication IP matches the submitted IP address, and (2) verifying the existence of a valid *Client Authz* record for the client. If both conditions are met, the CA proceeds with certificate issuance according to the standard ACME protocol. If the IP addresses do not match, the CA terminates the connection, as this mismatch may suggest that the ACME account information has been compromised. If no valid *Client Authz* record exists, the CA generates a new *Client Authz* object for the client, assigns it a "pending" status, and returns the object ID with challenges to the client.

The generation of challenges follows a structured approach: the CA first retrieves all valid *Authz* records associated with the account and randomly selects $n = \lceil \log(N + 1) \rceil$ domain names from these records, where $N$ denotes the total number of domains with valid *Authz* for the account. These $n$ domains serve as the foundation for new challenges, which the client must complete to re-establish domain ownership.

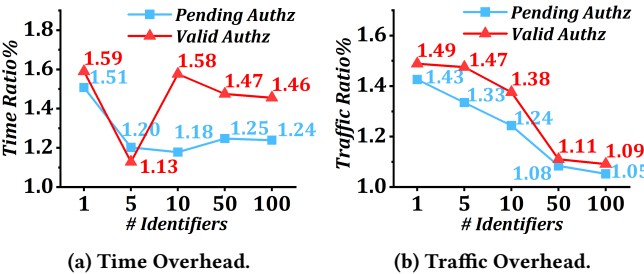

**(a) Time Overhead.**            **(b) Traffic Overhead.**

**Figure 6: ACME++ Overhead on CA Server Simulation Result in Worst Case Scenarios.**

ACME++ also introduces a new endpoint, *newClientAuthz*, to the ACME CA server Services (as detailed in Table 1). If the client proceeds to client verification, it sends a POST-as-GET request to /acme/client-authz/<authz_id> to access the *Client Authz* object, where <authz_id> corresponds to the ID of the *Client Authz* object. The client then completes the required challenges using either the HTTP-01 or DNS-01 challenge types. Upon successfully completing all challenges, the client is verified, and the CA updates the status of the *Client Authz* object to "valid." At this point, the CA may proceed with certificate issuance. Additionally, the *Client Authz* record is cached for 30 days, allowing the client to request further certificates within this period without needing re-verification.

In summary, the core principle of ACME++ is that if the client fails to complete the challenges, it signifies an inability to prove domain ownership, prompting the CA to terminate the issuance process. This mechanism prevents exploitation of the ACME *Authz* cache vulnerability, as discussed in Section 3.

### 4.4 Overhead Evaluation

In this section, we evaluate the ACME++ protocol design in terms of CA server overhead.

*4.4.1 Setup.* We implemented the updates for both ACME and ACME++ in Python, covering the CA server and the ACME client, following the RFC documentation. The implementation simulates the complete ACME protocol message exchange process, excluding certain checks and signature verification required in production environments. We configured the CA server to directly verify challenges on the client using the HTTP-01 challenge type. To simulate real-world certificate issuance between a CA and a client, we deployed the client and the server on separate hosts across different ISP networks.

*4.4.2 Overhead Analysis.* **Normal Case.** In typical use cases, such as a web server with a fixed public IP address that periodically requests certificates, the overhead remains minimal due to the 30-day caching of the *Client Authz* record. Consequently, both types of overhead occur infrequently, maintaining efficiency and minimizing traffic between the client and the CA. Additionally, the logarithmic selection of domain challenges greatly reduces challenge overhead in large-scale scenarios. For example, if a client manages $N = 10,000$ domains, it would only need to complete $n = \lceil \log(10,000 + 1) \rceil = 14$ challenges, representing just 0.14% of the total domains.

**Worst Case.** When clients operate with dynamic IP addresses (e.g.,

within a campus network), *Client Authz* validation may fail with each certificate request, necessitating re-verification and introducing overhead for each request. To evaluate this, we simulated two extreme scenarios to assess the certificate issuance overhead. In the first scenario, the client requests a certificate without any valid *Authz* records cached in the CA server (referred to as *Pending Authz*). Here, the client must undergo both the ACME++ client verification and the domain validation processes. The second scenario involves the client requesting a certificate with all valid *Authz* records cached in the CA server (referred to as *Valid Authz*), requiring only the client verification step.

We used two metrics for evaluation: traffic overhead, defined as the ratio of total bytes transmitted between the client and the CA server, and time overhead, measured as the ratio of the duration from the client's *newOrder* request to the final certificate issuance. The number of identifiers in each *newOrder* request ranged from 1, 5, 10, 50, to the maximum of 100 allowed by Let's Encrypt per certificate. To standardize the time for challenge completion, we assumed a fixed duration of 1 second per domain, with parallel processing for multiple domain challenges. Each process was repeated 10 times, and the average overhead ratios were calculated.

Figures 6a and 6b present the simulation results for time and traffic overhead in the ACME++ process. For time overhead, *Valid Authz* incurs an average additional ratio of approximately 40%, while *Pending Authz* experiences about 20%. The time overhead is more variable when the number of identifiers is small (e.g., 1 or 5), likely due to network variability. However, as the number of identifiers increases, the time overhead stabilizes, indicating more consistent performance. For traffic overhead, a clear decreasing trend emerges as the number of identifiers ($N_i$) increases. This trend suggests that the relative impact of challenge traffic diminishes with larger $N_i$, as the fixed communication costs are distributed over more identifiers, resulting in a lower per-identifier traffic overhead ratio.

In summary, even under conditions simulating maximum overhead, ACME++ maintains a time overhead below 60%, and traffic overhead consistently falls below 50%. These results demonstrate that ACME++ provides a scalable and efficient solution, even when managing a large number of identifiers.

### 4.5 Resilience to Potential Attacks

In this section, we demonstrate that the ACME++ protocol provides strong security against various attack scenarios.

*1. Client Authz Caching Attack.* Similar to the ACME *Authz* Cache Attack, in this scenario, the attacker possesses knowledge of the victim's ACME account key, UID, client IP, and client ID and attempts to exploit cached Client *Authz* records to obtain fraudulent certificates.

**Feasibility Analysis:** If the attacker uses their own IP address to communicate with the CA, the CA will detect a mismatch between the submitted client IP and the attacker's IP, causing the Client *Authz* verification to fail. If the attacker attempts to spoof the IP address to appear as the victim's, they would not receive responses from the CA unless they can control the victim's network traffic through methods like BGP prefix hijacking, which is highly unlikely. Thus, the attacker could only succeed if they possess both the

correct ACME account credentials and the capability to execute a BGP prefix hijack—an improbable combination.

*2. Brute Force Attack.* In this scenario, we assume the attacker has access to the victim's ACME key and UID, allowing them to communicate with the CA from the victim's IP address, but they lack the client ID.

**Feasibility Analysis:** If the client ID is composed of both uppercase and lowercase letters and digits, and its length is 8 characters, the complexity of a brute-force attack would be $62^8 \approx 2.18 \times 10^{14}$ (approximately 218 trillion possible combinations), rendering such an attack infeasible.

*3. Partial Domain Control Attack.* In this scenario, the attacker possesses the victim's ACME account key and UID and has control over some of the victim's domains. The attacker seeks to obtain a fraudulent certificate that includes other domains belonging to the victim.

**Feasibility Analysis:** During the account challenge process, the attacker must ensure that at least *n* domains under their control have valid *Authz* records to pass the challenge and proceed with the attack. Since the domains for the challenge are selected randomly, the attacker would need significant control over the victim's domain portfolio to meet this condition. At this point, the attacker would likely already possess substantial access to the victim's resources, making the attack scenario redundant.

# 5 Related Work

## 5.1 Web-PKI Attack Model

**Private Key Compromise.**
The Web-PKI ecosystem, based on cryptography and SSL certificates, relies on secure private key management for maintaining communication integrity. Early studies examined risks of private key leakage through public vulnerabilities, like the 2008 Debian OpenSSL flaw [37], allowing attackers to match public keys with known compromised ones, and the 2014 Heartbleed bug [38], which exposed keys by leaking server memory.

Subsequent work by Cangialosi *et al.* [9] revealed key reuse across certificates, including private keys accessible by third-party hosting providers like CDNs. More recently, Ma *et al.* [27] introduced *Certificate Invalidation Events*, where entities such as previous domain owners or CDN providers retained valid certificate keys despite changes in domain ownership.

While prior work has concentrated on SSL certificate private keys, our research focuses on compromised ACME account keys. We demonstrate that a compromised ACME account key has far-reaching consequences for Web-PKI, as it allows attackers to issue entirely new, valid fraudulent certificates using their key pairs, whereas compromised certificates can often be quickly revoked by their owners.

**Fraudulent Certificate Retrieval.**
To execute an MITM or phishing attack, attackers must first acquire a fraudulent certificate that matches the victim's identity. Prior research has developed several methods to obtain fraudulent certificates within the Web-PKI framework. These methods primarily target domain validation processes through either network-level or DNS-level attacks. In [5], Birge-Lee *et al.* demonstrate an attack

where BGP prefix hijacking allows an attacker to successfully complete domain validation and obtain a fraudulent certificate. Other notable attacks focus on DNS-based vulnerabilities. Borgolte *et al.* [7] present *CloudStrife*, which exploits dangling DNS records in cloud environments to obtain certificates for unallocated IP addresses. Similarly, Brandt *et al.* [8] introduces a DNS cache poisoning attack, wherein fragmented responses to a DNS resolver are manipulated to point the victim domain to an attacker-controlled IP address. Dai *et al.* [14] propose off-path attacks on nameservers to manipulate the domain validation process, forcing the CA to rely on attacker-specified nameservers.

Unlike these approaches, which involve deceiving the CA into validating attacker-controlled domains, our attack circumvents the domain validation step entirely by exploiting the ACME *Authz* caching mechanism. Once the attacker possesses the victim's ACME account private key, they can directly communicate with the CA to request a fraudulent certificate without triggering any domain validation. This approach significantly reduces the attacker's overhead, eliminating the need for DNS or network-level manipulation, and lowers the barrier for attackers.

## 5.2 CT Information Leakage

In our attack model, CT logs provide a wealth of information, enabling attackers to identify associated domains. As CT logs continue to grow, researchers have increasingly raised concerns about potential information leakage and privacy violations [30, 32, 33].

Scheitle *et al.* [33] were among the first to provide a heuristic analysis of both the positive and negative aspects of CT. They highlighted how CT logs expose FQDNs and subdomains that would otherwise be hidden from public scans. Kales *et al.* [24] further expanded on this by identifying sensitive information such as usernames, email addresses, and business relationships that could be inferred from CT logs. More recently, Pletinckx *et al.* [30] demonstrated how attackers could leverage CT logs to identify domains that have ceased renewing their certificates, which are more likely to be vulnerable to known security vulnerabilities.

In our work, we harness the publicly available CT logs to identify domains associated with the victim's ACME account, thus expanding the attack surface. This approach further underscores the dual nature of CT logs, which, while designed to increase transparency and security, can also inadvertently assist attackers in reconnaissance efforts.

# 6 Conclusion

This paper exposed a critical vulnerability in the ACME protocol, allowing attackers to obtain fraudulent certificates by exploiting the *Authz* cache mechanism. We introduced the ACME *Authz* Cached attack and found that the world's largest CA, Let's Encrypt, is vulnerable to this attack. We proposed ACME++, an enhanced protocol that introduces *Client Authz* on a new Directory service to strengthen client verification. ACME++ mitigates the risk of unauthorized certificate issuance without altering the existing PKI infrastructure or imposing significant overhead on CAs. Our solution effectively prevents attackers from exploiting compromised ACME accounts while maintaining the efficiency of certificate reissuance, ensuring continued trust in the Web-PKI ecosystem.

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
