# OpenReview forum: "ACME++: Secure ACME Client Verification for Web-PKI"
_ACM.org/TheWebConf/2025/Conference — WWW 2025 Oral_

### Official Review · Reviewer_SW7j · 2024-11-25

**Novelty:** 5
**Technical Quality:** 5

**Review:**

# Review
## Summary
Thank you for your submission. It was a very interesting and thought-provoking read. You touched on very relevant material.

The paper identifies a potential attack on the ACME protocols used by CA's in WebPKI by exploiting the domain validation cache. The work identified and described a possible attack and then proposed modifications to the protocol (ACME++) to mitigate the attack.
## Strengths and weaknesses

### Strengths
The paper's writing is straightforward to understand, and the associated figures clearly demonstrate the required information (the protocols and attack methods). There were no sections where I felt the details were unclear or raised questions. Thank you.

The paper was well-motivated and situated in the web domain.

The description of the attack methodology was precise and repeatable while still being easily understandable.

The analysis of searching for associated domains in the CT logs was fascinating to me. The methodology was well described, and the analysis had a lot of thought put into it.

The proposed modification in ACME++ addresses/prevents the attack quite neatly.
### Weaknesses
My primary concern with the paper is the viability of the proposed attack vector. The first assumption is that the attacker can access the account credentials. This is a strong assumption; if this is the case, then the question is how. If it is, as the paper proposed, through gaining access to the victim server, then the attacker potentially has far greater powers than the threat model assumes.

For argument's sake, assume that the attacker has gained RCE or full access to the victim's server and, through this, has found the account credentials and launched the attack. ACME++'s security relies on the fact that the attacker cannot send messages from the original IP to the ACME++ endpoints and that they do not have knowledge of the client ID. These two assumptions are weak and can be trivially argued to be false.

First, consider the IP. An attacker with RCE or machine access powers could easily send requests from the compromised server. The discussions of BGP protecting the IP don't provide any value if the attacker has moderate access to the server or other machines that may share the same IP.

Second, the client ID. It is argued that the attacker would not know the client ID. Again, presuming access to the server, unless the victim was careful, the client ID would be easily accessible to the attacker. If the attacker can get the account credentials, what prevents them from just as easily acquiring the client ID? It has to be stored somewhere, and by your threat model, it could be extracted.

The attack vector seems to boil down to the `Account UID` being incorrectly generated by the CA's (only Let's Encrypt -- what do the others do). The simple solution seems to be for the CA's to correctly follow the recommendations in the RFC spec of the ACME protocol and treat the account UID's as 'cryptographically secure.' Then, the brute-forcing approach is entirely infeasible. Even so, if the UID is incremental, a competent CA could easily identify the brute-forcing traffic and take corrective actions (rate limiting, firewalls, general traffic monitoring). While the CAs really should follow the RFC recommendations, the paper misrepresents the feasibility of determining the UID if it does not.

It seems that the client ID in ACME++ is just the Account UID, but this time, it has been 'done right'—or at least, it is the client's responsibility to do it right.

Considering all this, adding a 50% overhead to the protocol seems excessive.

On another note, assuming the authors believe the attack vector to be viable, the lack of responsible disclosure to the affected CAs is concerning. It seems to me that the researchers are responsible for notifying the CAs of the potential security impacts of their poor choices of `Account UIDs`.

### Originality / Significance
While the work is novel and original, I am not convinced that the proposed attack vector is significant or viable.

**Questions:**

# Questions
Thank you for your paper. I truly did find it to be a very engaging and enjoyable read.

How does your threat model account for the attacker being able to capture the account credentials but not the ACME++ client ID?

Does your threat model not account for the attacker gaining enough access to the victim's server to send malicious queries from the victim's IP?

What was the reasoning for not engaging in a responsible disclosure?

I think this paper is solid, and I am open to updating my review based on your feedback and clarifications.

**Ethics Review Description:**

The authors claim to identify a potential attack vector faced by CAs (the name Let's Encrypt as being vulnerable) but do not discuss responsible disclosure to the affected CAs in the ethics considerations section (3.5). I am not convinced of the attack vector, but I am still concerned about the lack of disclosure.

**Ethics Review Flag:**

Yes

**Reviewer Confidence:**

3: The reviewer is confident but not certain that the evaluation is correct

**Scope:**

4: The work is relevant to the Web and to the track, and is of broad interest to the community

---

### Official Review · Reviewer_9qzQ · 2024-11-30

**Novelty:** 5
**Technical Quality:** 4

**Review:**

This paper introduces the ACME Authz cache attack and proposes an enhanced protocol named ACME++. I appreciate this paper; however, there are a few concerns:
1.	Your title is "ACME++," yet the majority of the paper focuses on describing the ACME Authz cache attack, while the introduction of ACME++ takes up only a small portion. I believe this makes the emphasis unclear.
2.	You claim that ACME++ “preserving the existing Web-PKI infrastructure and compatibility with current CA configurations,” but I did not find evidence of compatibility demonstrated in the text or the experimental evaluation.
3.	The experimental evaluation of ACME++ is somewhat superficial and lacks data to support its claims. Could you expand the experiments to more comprehensively showcase the results?
4.	The assumptions about the attacker’s capabilities in the Threat Model are overly simplified, and some attack scenarios rely on obtaining the victim's ACME account credentials.

**Questions:**

1.	Can you expand the experiments?
2.	Can you elaborate on the demonstration of compatibility?

**Reviewer Confidence:**

3: The reviewer is confident but not certain that the evaluation is correct

**Scope:**

3: The work is somewhat relevant to the Web and to the track, and is of narrow interest to a sub-community

---

### Official Review · Reviewer_Gzyq · 2024-12-02

**Novelty:** 5
**Technical Quality:** 7

**Review:**

The work is well-structured, progressing logically from recognizing a problem to offering and evaluating solutions. The approach and experimentation are detailed. The paper is clear and concise, with well-explained technical details and figures to support the narrative. Key concepts like the ACME protocol and the proposed enhancements (ACME++) are adequately described for readers familiar with the domain. In terms of originality, identifying the ACME Authz Cache Attack is novel and highlights a critical vulnerability in a widely adopted protocol. The proposed ACME++ protocol presents an innovative yet practical solution.  While this novelty is evident, it builds heavily on existing protocols and does not introduce groundbreaking cryptographic or system-level advancements.

This change may face adoption challenges since implementing ACME++ requires changes to existing CA workflows, which may hinder widespread adoption.

**Questions:**

1. Could you elaborate on the risks associated with using Certificate Transparency (CT) logs for reconnaissance? Are there strategies to mitigate this?
2. How resilient is ACME to brute-force attempts on client IDs?

**Reviewer Confidence:**

3: The reviewer is confident but not certain that the evaluation is correct

**Scope:**

4: The work is relevant to the Web and to the track, and is of broad interest to the community

---

### Official Review · Reviewer_rWzs · 2024-12-02

**Novelty:** 5
**Technical Quality:** 4

**Review:**

Pros:
- Clear and detailed technical explanations
- Well-organized structure

Cons:
- Unclear threat model
- Lack of strong justification for the attack assumptions

Detailed Comments:
Thank you for the submission. The paper proposes the ACME Authz Cache Attack, which aims to exploit leaked credentials in order to bypass the authentication process on Certificate Authority (CA) servers and request certificates without proving domain ownership. While the attack seems promising, I believe the threat model is not fully justified, as outlined below.

1. Attacker Abilities in Section 3.1:
The authors state that "in our threat model, the attacker only has a standard Internet connection that can scan web servers to uncover vulnerabilities and domain services." However, in the Attack Assumptions section, it is mentioned that "The attacker obtains the victim’s ACME account credentials by exploiting a server-side vulnerability." This is an extremely strong assumption, and I’m unclear on the purpose of defining the attacker’s abilities in a separate section when such a strong assumption is already made.

2. Unjustified Assumption of ACME Account Credential Leakage:
The assumption that ACME account credentials are leaked is not adequately justified. For example, consider the Path Traversal Vulnerability discussed in the paper. Certbot takes care to set private key permissions to 600, which restricts access to the file to the owner or root. Even if we assume a path traversal vulnerability exists, a web server process still wouldn't be able to access the private key, as it doesn't have sufficient permissions. Given that Certbot is used as the example in this paper, I am left wondering under what specific scenarios the assumption of credential leakage is reasonable.

**Questions:**

1. Given that Certbot (the example used) sets private key permissions to 600, which restricts access to the key even in the case of a path traversal vulnerability, could you clarify under what specific scenarios this assumption becomes plausible? What conditions would make the leakage of these credentials feasible?

2. In Section 3.1, you mention that the attacker has only a standard Internet connection for scanning vulnerabilities, yet in the Attack Assumptions section, you propose that the attacker can exploit a server-side vulnerability to gain access to sensitive credentials. How do you reconcile these two parts of the threat model? Is the attacker’s ability to exploit a vulnerability supposed to be part of the assumed attack, or is it an unintended contradiction in your model?

**Reviewer Confidence:**

4: The reviewer is certain that the evaluation is correct and very familiar with the relevant literature

**Scope:**

4: The work is relevant to the Web and to the track, and is of broad interest to the community

---

### Official Review · Reviewer_ZNdz · 2024-12-02

**Novelty:** 6
**Technical Quality:** 6

**Review:**

# Summary
This paper introduces the ACME Authz Cache Attack, an attack that exploits a vulnerability allowing attackers to obtain fake certificates by leveraging compromised ACME account credentials and cached authorization records without domain revalidation. These attacks can propagate damage to other domains associated with the same account via CT logs. The study confirms that Let’s Encrypt, the largest CA, is vulnerable to this attack. To mitigate this, the paper proposes ACME++, which enhances ACME security by binding a client’s IP address and a unique identifier to the ACME account and requiring revalidation for every certificate request.

# Pros
- Effectively introduces an attack exploiting vulnerabilities in popular protocols and demonstrates its criticality through real-world evaluation.
- The paper follows a logical flow, transitioning seamlessly from the introduction of the attack to proposing mitigation techniques.

# Cons
- Some terms are used before being clearly defined, making them difficult to understand (e.g., "associated domains").

**Questions:**

- In Section 4.5.1, it is stated that a BGP prefix hijacking attack is unlikely to occur in practice. However, the paper cites [5], which demonstrates such an attack. Could you explain the relationship between BGP prefix hijacking attacks and the proposed technique? If the proposed technique is unrelated to this type of attack, wouldn’t it be more appropriate to exclude this attack from the threat model?

**Reviewer Confidence:**

2: The reviewer is willing to defend the evaluation, but it is likely that the reviewer did not understand parts of the paper

**Scope:**

4: The work is relevant to the Web and to the track, and is of broad interest to the community